# LoRA Unleashed: Effortlessly Advancing from Low to Arbitrary Rank

## ABSTRACT

Low-Rank Adaptation (LoRA) has emerged as a prominent technique for fine-tuning large foundation models, facilitating a reduction in trainable parameters through the utilization of low-rank matrices to represent weight changes $\mathbf{A}$ and $\mathbf{B}$ (*i.e.*, $\Delta \mathbf{W} = \mathbf{BA}$). Although LoRA has demonstrated considerable success, its expressiveness is inherently limited by the constrained capacity of its low-rank structure. To ameliorate this limitation, we introduce Fourier-based Flexible Rank Adaptation (FoRA), which harnesses the robust expressiveness of the Fourier basis to re-parameterize $\mathbf{A}$ and $\mathbf{B}$ from a sparse spectral subspace. Utilizing FoRA, adaptation matrices can overcome conventional rank limitations, achieving up to a 15x reduction in the parameter budget. We illustrate that FoRA achieves an optimal balance of efficiency and performance across various tasks, including natural language understanding, mathematical reasoning, commonsense reasoning, and image classification. Our codes are available at https://anonymous.4open.science/r/FoRA-0E9C.

## 1 INTRODUCTION

In recent years, Large Foundation Models (LFMs), have showcased exceptional generalization capabilities, greatly improving performance in a wide array of tasks across natural language processing (NLP) (Brown et al., 2020; Touvron et al., 2023a), computer vision (CV) (Radford et al., 2021; Kirillov et al., 2023), and other fields (Azad et al., 2023). Typically, adapting these general models for specific downstream tasks requires full fine-tuning, which involves retraining all model parameters and can pose significant challenges, particularly in resource-limited environments. To address this issue, Parameter-efficient fine-tuning (PEFT) techniques (Mangrulkar et al., 2022), have been developed, offering more feasible alternatives. Among these, Low-Rank Adaptation (LoRA) (Hu et al., 2021), which decomposes the weight changes into the product of two low-rank matrices $\mathbf{A}$ and $\mathbf{B}$, has stood out for its effectiveness and simplicity.

Despite its success, LoRA's reliance on low-rank structures can limit its expressive potential. Theoretically, the expressive capacity of LoRA is constrained by the ranks of $\mathbf{A}$ and $\mathbf{B}$ (Zeng & Lee, 2023). Therefore, more complex downstream tasks inherently necessitate higher ranks (Hu et al., 2023; Biderman et al., 2024; Gao et al., 2024). To elucidate the significance of rank configurations in practical applications, we delve into their effect on LoRA's performance across various tasks and present the corresponding observations in Figure 1. As depicted, while different tasks exhibit varying sensitivities to rank, most demonstrate improved performance as the rank increases, with performance peaking at higher ranks (*i.e.,* no less than $2^6$). This pattern aligns with the behavior of LoRA when applied to the LLaMA family, where high ranks yield clear improvements (Biderman et al., 2024). However, adapting LoRA to higher ranks inevitably engenders larger trainable parameter sizes, thereby imposing considerable overhead. Hence, a question is naturally raised:

*How can we unleash the rank-bounded potential of LoRA*
*while still residing in the low-parameter jail?*

This question aligns closely with the principles of sparse learning (Han et al., 2015a), which aim to preserve expressive information while necessitating fewer learnable parameters. Despite the success of its predominant pruning techniques (Han et al., 2015b; Frankle & Carbin, 2018), determining which modules to prune often requires complex strategies (Zhang et al., 2022). In contrast, classical

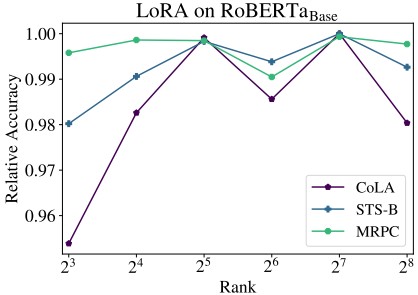 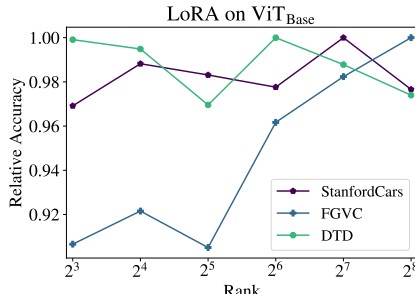

Figure 1: LoRA applied to RoBERTa$_{\text{BASE}}$ and ViT$_{\text{BASE}}$ under varying rank configurations. All experiments followed a comprehensive hyperparameter search. The reported relative accuracy, averaged over five distinct random seeds, reflects performance compared to the best results of each task. Notably, performance peaks at higher rank configurations across all tasks.

data compression techniques, such as linear projection (Dony & Haykin, 1995), fractal compression (Cochran et al., 1996), and spectral transformations (Reddy & Murthy, 1986), can be applied directly to weight matrices, providing a simpler yet effective alternative. Among these, the Fourier basis, which enables high-quality data recovery from sparse spectral information (Rudelson & Vershynin, 2006; Duarte & Baraniuk, 2013; Vlaardingerbroek & Boer, 2013), stands out as a promising tool for sparse learning. We refer our readers to Section 4.5 for a more in-depth empirical analysis.

To this end, we propose Fourier-based Flexible Rank Adaptation (FoRA), which leverages Fourier bases to re-parameterize adaptation matrices **A** and **B** as the spatial equivalents of sparse spectral components. Specifically, FoRA learns only $n$ spectral components at the predefined spectral locations, which are shared among all adaptation matrices. Then, inverse Fast Fourier Transform is applied to derive these adaptation matrices in the spatial space. It is important to note that the use of a fixed quantity of spectral components enables FoRA to facilitate the adjustment of **A** and **B** from lower to potentially unbounded ranks at fixed parameter cost, thus ensuring significant expressiveness within a constrained parameter scope. In summary, our contributions are as follows:

- Given the rank-dependent performance of LoRA, we introduce FoRA, a novel PEFT method that enhances LoRA with Fourier-based compression, maximizing its potential while minimizing the parameter overhead.

- FoRA consistently yields comparable or better performance with up to 15x fewer trainable parameters than LoRA on various tasks, from language to vision domains and across backbones in different scales, including RoBERTa, ViT and LLaMA.

- A thorough analysis is conducted to further substantiate FoRA as a parameter-efficient alternative that replicates LoRA's potential across different configurations.

## 2 RELATED WORKS

### 2.1 PARAMETER-EFFICIENT FINE-TUNING

Fine-tuning large pre-trained language models is crucial for improving NLP tasks. However, updating all model parameters is computationally intensive and storage-demanding for models like GPT-3 (Brown et al., 2020) and LLaMA (Touvron et al., 2023a). Parameter-efficient fine-tuning (PEFT) methods address these issues by updating fewer parameters or adding lightweight modules.

One prominent approach in PEFT is the use of adapters — small bottleneck layers inserted within each layer of a pre-trained model (Houlsby et al., 2019; Pfeiffer et al., 2020; Karimi Mahabadi et al., 2021; He et al., 2021). Houlsby et al. (2019) introduced adapters that enable task-specific adaptation while keeping the original model weights fixed. Building upon this, Pfeiffer et al. (2020) proposed a modular adapter framework that facilitates multi-task transfer. To further optimize parameter efficiency, Karimi Mahabadi et al. (2021) reduced the number of parameters by employing parameter

sharing and low-rank approximations within adapters. Another line of research involves prompt tuning, which modifies the input embeddings to guide the model toward specific tasks (Lester et al., 2021; Liu et al., 2021; Li & Liang, 2021; Chen et al., 2023a). Lester et al. (2021) optimized continuous prompt embeddings while keeping the language model's parameters fixed, demonstrating the effectiveness of prompt tuning for task adaptation. Similarly, Prefix-Tuning (Li & Liang, 2021) prepends trainable vectors to the input of each transformer layer without altering the model architecture, effectively steering the model toward desired behaviors with minimal parameter updates.

While these methods exhibit high efficiency and preserve the originality of the pre-trained model, they inevitably introduce higher inference costs due to additional modules or modifications required during deployment. In contrast, LoRA (Hu et al., 2021) and its variants (Zhang et al., 2023a; Bałazy et al., 2024; Li et al., 2024; Liu et al., 2024; Nikdan et al., 2024; Gao et al., 2024) inject trainable low-rank matrix decomposition into transformer layers. This approach not only reduces the number of trainable parameters but also allows for merging these decompositions with the original model weights, thereby avoiding increased inference burdens. However, the expressiveness of low-rank adaptation methods like LoRA is often bounded by the chosen rank (Zeng & Lee, 2023). To address this limitation, Kopiczko et al. (2023) and Jiang et al. (2024) explored high-rank adaptations through projection matrices, aiming to enhance expressive capacity without significantly increasing parameter counts. Despite these advances, our empirical experiments indicate that while LoRA's performance may peak at certain high-rank configurations, increasing the rank beyond this point does not necessarily lead to better results.

## 2.2 SPARSE LEARNING

Sparse neural networks exploit the fact that many weights in over-parameterized models can be pruned with minimal impact on performance (Han et al., 2015b; Lee et al., 2018; Frankle & Carbin, 2018; Wang et al., 2020; Liu et al., 2022; Frantar & Alistarh, 2023). Techniques such as magnitude pruning (Han et al., 2015a) remove weights with small magnitudes, effectively reducing model size. Dynamic sparsity methods (Mocanu et al., 2018; Zhang et al., 2022; Chen et al., 2023b) adjust the sparsity patterns during training, allowing the network to discover efficient architectures on the fly.

Another innovative approach is learning in transformed domains like the sparse Fourier space. By representing weight matrices in the frequency domain using Fourier transforms, neural networks can exploit the sparsity inherent in the frequency representation of the data (Yang & Xie, 2016; Chen et al., 2016). This allows for efficient compression by retaining only the significant frequency components and discarding the less important ones. Learning in the sparse Fourier space facilitates the development of compact models that effectively capture essential patterns with fewer parameters.

## 3 METHODOLOGY

### 3.1 BACKGROUND

**Low-Rank Adaptation (LoRA)** LoRA (Hu et al., 2021) proposes to use the product of two low-rank matrices to update the pre-trained weights incrementally. Let $\mathbf{W}' \in \mathbb{R}^{d_1 \times d_2}$ deotes the fine-tuned weight, $\mathbf{W}_0 \in \mathbb{R}^{d_1 \times d_2}$ the pre-trained weight, and $\Delta\mathbf{W} \in \mathbb{R}^{d_1 \times d_2}$ the change in weight. LoRA models this change $\Delta\mathbf{W}$ through a low-rank decomposition:

$$\mathbf{W}' = \mathbf{W}_0 + \Delta\mathbf{W} = W_0 + \mathbf{B}\mathbf{A}, \tag{1}$$

where $\mathbf{W}_0$ is kept unchanged during fine-tuing. The matrices $\mathbf{A} \in \mathbb{R}^{r \times d_2}$ and $\mathbf{B} \in \mathbb{R}^{d_1 \times r}$ represents the learnable low-rank matrices with the rank $r \ll \{d_1, d_2\}$. Typically, $\mathbf{A}$ adopts Kaiming uniform initialization (He et al., 2015) while $\mathbf{B}$ is initialized to zero at the start of the training process.

In the following parts, we present Fourier-based Flexible Rank Adaptation (FoRA), which re-parameterizes adaptation matrices of LoRA by applying inverse Fast Fourier Transform on sparse spectral coefficients. The overall framework is presented in Figure 2.

### 3.2 FOURIER-BASED FLEXIBLE RANK ADAPTATION

As stated previously, our goal is to re-parameterize $\mathbf{A}$ and $\mathbf{B}$ with fewer parameters while maintaining strong expressiveness, which aligns closely with the foundational principle of sparse learning.

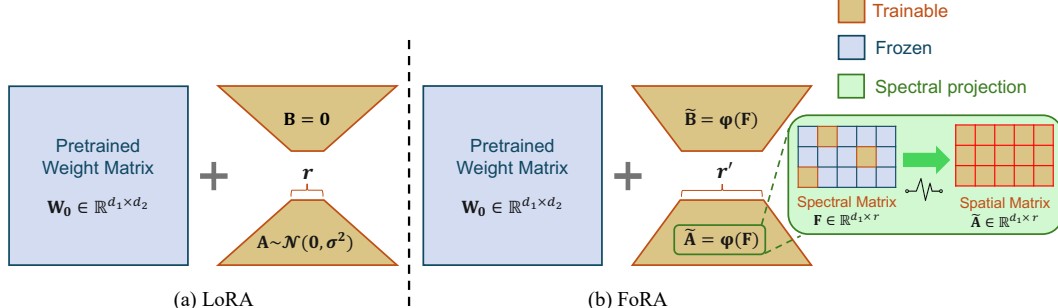

(a) LoRA  (b) FoRA

Figure 2: An overview of the schematic comparison between LoRA and our proposed FoRA. While LoRA necessitates training all elements in the low-rank matrices $\mathbf{A}$ and $\mathbf{B}$, FoRA re-parameterizes these matrices from a sparse spectral subspace (highlighted in green). Our approach enables flexible rank adjustment while training fixed and sparse components. In both cases, low-rank matrices can be merged into the original weights matrix $\mathbf{W}_0$, ensuring no additional latency is introduced.

Upon revisiting prior successes, we resort to the Fourier basis, known for its robust expressiveness (Candès et al., 2006; Baraniuk, 2007).

Essentially, our approach centers on re-parameterizing the adaptation matrices, termed $\tilde{\mathbf{A}} \in \mathbb{R}^{r \times d_2}$ and $\tilde{\mathbf{B}} \in \mathbb{R}^{d_1 \times r}$, as the spatial recovery of sparse spectral coefficients, while retaining LoRA's update schema:

$$\mathbf{W}' = \mathbf{W}_0 + \Delta\mathbf{W} = \mathbf{W}_0 + \tilde{\mathbf{B}}\tilde{\mathbf{A}}. \qquad (2)$$

To accomplish this, we start by randomly initializing a 2D index matrix $\mathbf{L} \in \mathbb{R}^{2 \times n}$ to specify spectral locations for all low-rank matrices. To derive $\tilde{\mathbf{A}}$, we then define $n$ learnable spectral coefficient $\mathbf{s} \in \mathbb{R}^n$. Using these indices and coefficients, we construct the sparse spectral matrix $\mathbf{F} \in \mathbb{R}^{r_1 \times d}$ and compute its spatial counterpart $\mathbf{S} \in \mathbb{R}^{r_1 \times d}$ via the inverse Fast Fourier Transform:

$$\mathbf{S}_{p,q} = \frac{1}{rd_2} \sum_{j=0}^{r-1} \sum_{k=0}^{d_2-1} \mathbf{F}_{j,k} e^{i2\pi(\frac{j}{r}p + \frac{k}{d_2}q)}, \qquad (3)$$

where $i$ denotes the imaginary unit. In particular, $\mathbf{F}_{j,k} = \mathbf{s}_p$ if $(j,k) = \mathbf{L}_{:,p}$ and $\mathbf{F}_{j,k} = 0$ otherwise. The Fourier-based re-parameterized matrix $\tilde{\mathbf{A}}$ is then defined as the real part of the complex matrix $\mathbf{S}$ as

$$\tilde{\mathbf{A}} = \text{Re}[\mathbf{S}]. \qquad (4)$$

The adaptation matrix $\tilde{\mathbf{B}}$ is obtained by applying the identical procedure as above.

In this setup, FoRA can be easily integrated as a plug-in by replacing the LoRA linear module with the FoRA linear module in a single line of code, requiring no additional modifications, as outlined in Algorithm 1 in the Appendix. Moreover, despite learning only a fixed number of spectral components, the high expressiveness of the Fourier basis allows FoRA to represent informative matrices with ranks that range from low to very high values. This flexibility enables FoRA to replicate LoRA's potential, even within a constrained parameter space.

## 3.3 DISCUSSION

**Initialization strategies.** Matrix initialization with consistent variance (Glorot & Bengio, 2010) is crucial for maintaining numerical stability and accelerating convergence. However, unlike LoRA, directly initializing the spectral space in FoRA can lead to suboptimal variance in spatial space due to the involvement of the Fourier transform. To facilitate efficient training, for matrix $\tilde{\mathbf{A}}$, we first employ Xavier (Glorot & Bengio, 2010) or Kaiming initialization (He et al., 2015) to its spectral coefficients $\mathbf{s}$ and a spatial auxiliary matrix $\mathbf{A}' \in \mathbb{R}^{r \times d_2}$. Next, we scale $\mathbf{s}$ by $\text{Var}(\mathbf{A}')/\text{Var}(\tilde{\mathbf{A}})$ to approximate consistent variance. In contrast, matrix $\tilde{\mathbf{B}}$ is initialized to zeros following the standard practice of LoRA (Hu et al., 2021). We employ Kaiming initialization by default unless specially stated.

**Comparison to LoRA's variants.** Recent parameter-efficient variants of LoRA (Kopiczko et al., 2023; Renduchintala et al., 2023; Li et al., 2024) have demonstrated competitive performance by adapting at higher ranks through the use of simple linear projections. However, their strategies for sparse learning, which essentially involve a collection of learnable scaling transformations, suffer from limited expressiveness. To remedy this issue, FoRA leverages the more efficient and expressive Fourier transform for matrix re-parameterization, striking a balance between performance and efficiency. Compared with them, FoRA consistently provides enhanced representational expressiveness while allowing flexible rank adaptation with fixed cost. Further details of the empirical analysis are provided in Section 4.5.

# 4 EXPERIMENTS

In this section, we present a series of experiments to demonstrate the effectiveness of FoRA across diverse tasks, including language and image domains. We begin by evaluating FoRA through fine-tuning RoBERTa on the GLUE benchmark. Next, we focus on instruction tuning within the LLaMA family. Following this, we assess FoRA's performance by fine-tuning Vision Transformers for image classification. Finally, we provide an in-depth analysis of FoRA's capabilities.

**Baselines.** We evaluate FoRA against three groups of baselines. The first group follows the classical fine-tuning paradigm, which includes **Full Fine-tuning (FF)** and **BitFit** (Zaken et al., 2021) where only bias vectors are fine-tuned. The second group is adapter-tuning, covering **Adpt$^{\text{H}}$** (Houlsby et al., 2019), **Adpt$^{\text{P}}$** (Pfeiffer et al., 2020), **Adpt$^{\text{R}}$** (He et al., 2021). The third group is the most prevalent low-rank adaptation and its variants, including **LoRA** (Hu et al., 2021), **VeRA** (Kopiczko et al., 2023), **FourierFT** (Gao et al., 2024), **DoRA** (Liu et al., 2024).

## 4.1 GLUE BENCHMARK

We evaluate FoRA on the General Language Understanding Evaluation (GLUE) benchmark (Wang, 2018), a sequence classification benchmark for natural language understanding (NLU) which covers domains such as sentiment classification and natural language inference. We employ the pre-trained RoBERTa$_{\text{BASE}}$ and RoBERTa$_{\text{LARGE}}$ (Liu, 2019) as the foundation models for fine-tuning.

Our experimental setup closely follows (Hu et al., 2021), involving fine-tuning only the query and value weights in each transformer block and fully fine-tuning the classification head. For our method, we randomly sample $n = \{250, 500\}$ trainable spectral coefficients per low-rank matrix, which we denote as FoRA$^{\dagger}$ and FoRA, respectively. We adopt the baseline hyperparameters from their original papers. For our approaches, we apply random search (Bergstra et al., 2013) to optimize the learning rates and matrix rank. For comprehensiveness, we report the median performance across 5 random seed trials, selecting the best epoch for each run. Additionally, we report the number of trainable parameters in the fine-tuned layers, excluding the classification head, as suggested by (Hu et al., 2021; Kopiczko et al., 2023). Further specifics are provided in Table 6 in the Appendix.

**Results.** As highlighted in Table 1, FoRA generally delivers better or on-par performance compared with baseline methods, while adapting at higher ranks with extremely lower budget. Notably, under the same parameter constraints, FoRA demonstrates improved performance over FourierFT. The performance gains are more pronounced with the RoBERTa$_{\text{LARGE}}$ model. Specifically, FoRA$^{\dagger}$ not only surpasses adapter tuning by a clear margin but also matches the performance of LoRA, despite requiring 30 times fewer trainable parameters. These results demonstrate that FoRA strikes an effective balance between unleashing LoRA's rank-bounded potential and parameter efficiency.

## 4.2 MATHEMATICAL REASONING

Instruction tuning involves fine-tuning a language model on a collection of paired prompts and responses (Ouyang et al., 2022). To evaluate the effectiveness of FoRA, we first apply it to LLaMA2$_{\text{7B/13B}}$ (Touvron et al., 2023b) and LLaMA3$_{\text{8B}}$ (Dubey et al., 2024) for mathematical reasoning tasks.

This evaluation uses two challenging benchmarks: GSM8K (Cobbe et al., 2021) and MATH (Hendrycks et al., 2020). Both datasets consist of multi-step problems that require chain-

Table 1: Fine-tuning performance of the pre-trained RoBERTa$_\text{BASE}$ and RoBERTa$_\text{LARGE}$ models with different methods on the GLUE benchmark. We report Matthew's correlation coefficient for CoLA, Pearson correlation coefficient for STS-B, and accuracy for all the remaining tasks. The best results for each dataset are highlighted in **bold**. FoRA$^\dagger$: the lightweight version of FoRA.

| | Methods | # Trainable Parameters | SST-2 | MRPC | CoLA | QNLI | RTE | STS-B | Avg. |
|---|---|---|---|---|---|---|---|---|---|
| BASE | FF | 125M | 94.8 | 90.2 | 63.6 | 92.8 | 78.7 | 91.2 | 85.2 |
| | BitFit | 0.1M | 93.7 | **92.7** | 62.0 | 91.8 | **81.5** | 90.8 | **85.4** |
| | LoRA | 0.3M | **95.1**$_{\pm0.2}$ | 89.7$_{\pm0.7}$ | 63.4$_{\pm1.2}$ | **93.3**$_{\pm0.3}$ | 78.8$_{\pm0.5}$ | **91.5**$_{\pm0.2}$ | 85.3 |
| | VeRA | 0.043M | 94.6$_{\pm0.1}$ | 89.5$_{\pm0.5}$ | **65.6**$_{\pm0.8}$ | 91.8$_{\pm0.2}$ | 78.7$_{\pm0.7}$ | 90.7$_{\pm0.2}$ | 85.2 |
| | FourierFT | 0.024M | 94.2$_{\pm0.3}$ | 90.0$_{\pm0.8}$ | 63.8$_{\pm1.6}$ | 92.2$_{\pm0.1}$ | 79.1$_{\pm0.5}$ | 90.8$_{\pm0.2}$ | 85.0 |
| | **FoRA**$^\dagger$ | **0.012M** | 94.3$_{\pm0.3}$ | 89.7$_{\pm0.2}$ | 62.6$_{\pm1.6}$ | 92.4$_{\pm0.4}$ | 78.7$_{\pm2.6}$ | 90.0$_{\pm0.3}$ | 84.6 |
| | **FoRA** | **0.024M** | 94.7$_{\pm0.3}$ | 90.4$_{\pm0.5}$ | 64.6$_{\pm1.0}$ | 92.3$_{\pm0.1}$ | 79.4$_{\pm1.9}$ | 90.7$_{\pm0.2}$ | **85.4** |
| LARGE | Adpt$^\text{P}$ | 0.8M | **96.6**$_{\pm0.2}$ | 89.7$_{\pm1.2}$ | 67.8$_{\pm2.5}$ | **94.8**$_{\pm0.3}$ | 80.1$_{\pm2.9}$ | 91.9$_{\pm0.4}$ | 86.8 |
| | Adpt$^\text{H}$ | 0.8M | 96.3$_{\pm0.5}$ | 87.7$_{\pm1.7}$ | 66.3$_{\pm2.0}$ | 94.7$_{\pm0.2}$ | 72.9$_{\pm2.9}$ | 91.5$_{\pm0.5}$ | 84.9 |
| | LoRA | 0.8M | 96.2$_{\pm0.5}$ | 90.2$_{\pm1.0}$ | **68.2**$_{\pm1.9}$ | **94.8**$_{\pm0.3}$ | 85.2$_{\pm1.1}$ | **92.3**$_{\pm0.5}$ | 87.8 |
| | VeRA | 0.061M | 96.1$_{\pm0.1}$ | 90.9$_{\pm0.7}$ | 68.0$_{\pm0.8}$ | 94.4$_{\pm0.2}$ | 85.9$_{\pm0.7}$ | 91.7$_{\pm0.8}$ | 87.8 |
| | FourierFT | 0.048M | 96.0$_{\pm0.2}$ | 90.9$_{\pm0.3}$ | 67.1$_{\pm1.4}$ | 94.4$_{\pm0.4}$ | **87.4**$_{\pm1.6}$ | 91.9$_{\pm0.4}$ | 88.0 |
| | **FoRA**$^\dagger$ | **0.024M** | 96.1$_{\pm0.2}$ | 91.2$_{\pm1.0}$ | 66.5$_{\pm0.9}$ | 94.2$_{\pm0.5}$ | 86.6$_{\pm1.1}$ | 91.4$_{\pm0.2}$ | 87.7 |
| | **FoRA** | **0.048M** | 96.3$_{\pm0.1}$ | **91.4**$_{\pm1.0}$ | 68.0$_{\pm2.0}$ | 94.4$_{\pm0.3}$ | 87.0$_{\pm2.0}$ | 91.9$_{\pm0.4}$ | **88.2** |

of-thought reasoning (Wei et al., 2022) to reach the final answer, and they are framed as question-answering tasks using the same prompt template as in (Zhang et al., 2023b). Each method is fine-tuned on the respective training sets and evaluated on the testing sets, where we only evaluate the correctness of the final numeric answer.

In addition, FoRA only re-parameterizes the adaptation matrix with Fourier transform, thus allowing it to be adapted to other LoRA variants. To test the adaptability, we select DoRA, where the directional component of the decomposed weight is learnable, and apply FoRA to the directional update, resulting in a combination called DFoRA. We use $n = 30000$ learnable spectral coefficients for LLaMA2$_\text{13B}$ and $n = 20000$ for the rest. To ensure a fair comparison, we fine-tuned the models following the setup suggested in (Hu et al., 2023; Liu et al., 2024), keeping the baseline models at a fixed rank of $r = 32$ while experimenting with different learning rates. In contrast, for our approaches, we optimize both the learning rates and matrix ranks. For comprehensiveness, we consider two scenarios: (1) a standard single training pass and (2) extended training over three epochs, reporting the best results for each (Nikdan et al., 2024). A more detailed setup is provided in Table 7 in the Appendix.

Table 2: Comparison of LLaMA2$_\text{7B}$, LLaMA2$_\text{13B}$ and LLaMA3$_\text{8B}$ fine-tuned on mathematical benchmark datasets. Avg. denotes the average accuracy. The best results for each dataset are highlighted in **bold**.

| | | | GSM8K | MATH | Avg. | GSM8K | MATH | Avg. |
|---|---|---|---|---|---|---|---|---|
| | Methods | # Parameters | | 1 Epoch | | | Extended | |
| LLaMA2$_\text{7B}$ | LoRA | 16.8M | 27.07 | 4.35 | 15.71 | **38.53** | 5.70 | **22.12** |
| | DoRA | 17.0M | **28.20** | **4.55** | **16.38** | 38.06 | **6.05** | 22.06 |
| | **FoRA** | **2.56M** | 26.99 | 4.15 | 15.57 | 37.63 | 5.70 | 21.67 |
| | **DFoRA** | **2.82M** | 27.77 | 4.30 | 16.04 | 37.76 | 5.90 | 21.83 |
| LLaMA2$_\text{13B}$ | LoRA | 26.2M | 38.51 | 5.30 | 21.90 | 49.20 | 8.45 | 28.83 |
| | DoRA | 26.6M | 38.82 | 5.85 | 22.34 | 50.34 | **9.00** | 29.67 |
| | **FoRA** | **4.80M** | 37.54 | **6.20** | 21.87 | 48.98 | 8.65 | 28.81 |
| | **DFoRA** | **5.21M** | 39.58 | 5.55 | **22.56** | **50.49** | 8.90 | **29.70** |
| LLaMA3$_\text{8B}$ | LoRA | 13.6M | 53.16 | 18.95 | 36.06 | 62.45 | 21.25 | 41.85 |
| | DoRA | 13.8M | 54.28 | **20.55** | 37.42 | 62.55 | 22.20 | 42.38 |
| | **FoRA** | **2.56M** | 54.13 | 19.55 | 36.84 | **63.00** | 21.35 | 42.18 |
| | **DFoRA** | **2.72M** | **55.65** | 19.40 | **37.53** | 62.77 | **22.45** | **42.61** |

**Results.** The results in Table 2 show that FoRA and DFoRA achieve accuracy that closely matches or slightly surpasses baseline methods, even with over 5 times fewer trainable parameters, in both single-pass and extended training scenarios. Notably, DFoRA shows significant improvements over FoRA, highlighting the flexible adaptability of the FoRA framework. Our approaches are particularly effective with the more advanced LLaMA3$_{8B}$ model, indicating that FoRA is especially well-suited to the sophisticated post-training techniques used in the latest LLaMA family. Overall, these empirical observations underscore the effectiveness and strong compatibility of FoRA.

## 4.3 COMMONSENSE REASONING

For a comprehensive evaluation of instruction tuning, we further compare our methods with LoRA and DoRA on LLaMA$_{7B/13B}$ (Touvron et al., 2023a), LLaMA2$_{7B}$ (Touvron et al., 2023b), and LLaMA3$_{8B}$ (Dubey et al., 2024) for commonsense reasoning tasks.

These commonsense reasoning tasks are framed as multiple-choice questions across eight distinct datasets, including BoolQ (Clark et al., 2019), PIQA (Bisk et al., 2020), SIQA (Sap et al., 2019), HellaSwag (Zellers et al., 2019), WinoGrande (Sakaguchi et al., 2021), ARC-e, ARC-c (Clark et al., 2018), and OBQA (Mihaylov et al., 2018). Consistent with the approach in (Hu et al., 2023), we use the Commonsense170K dataset for training, which integrates the training sets of all eight datasets, while evaluations are conducted on the test sets of the individual datasets.

In our experiments, we set rank $r = 32$ for all models as suggested by (Liu et al., 2024). Given the complexity of the tasks, we use $n = 40000$ learnable spectral coefficients for LLaMA$_{13B}$ and $n = 30000$ for the rest. A detailed configuration setup is provided in Table 8 in the Appendix.

Table 3: Comparison of LLaMA$_{7B}$, LLaMA$_{13B}$, LLaMA2$_{7B}$ and LLaMA3$_{8B}$ against various methods on eight commonsense datasets. Results of all baseline methods are taken from (Liu et al., 2024). The best and runner-up models for each dataset are highlighted in **bold** and underline.

| | Methods | # Parameters | BoolQ | PIQA | SIQA | HellaS. | WinoG. | ARC-e | ARC-c | OBQA | Avg. |
|---|---|---|---|---|---|---|---|---|---|---|---|
| ChatGPT | — | — | 73.1 | 85.4 | 68.5 | 78.5 | 66.1 | 89.8 | 79.9 | 74.8 | 77.0 |
| LLaMA$_{7B}$ | Adpt$^H$ | 132M | 63.0 | 79.2 | 76.3 | 67.9 | 75.7 | 74.5 | 57.1 | 72.4 | 70.8 |
| | Adpt$^R$ | 239M | 67.9 | 76.4 | **78.8** | 69.8 | 78.9 | 73.7 | 57.3 | 75.2 | 72.2 |
| | LoRA | 55.7M | **68.9** | 80.7 | 77.4 | 78.1 | 78.8 | 77.8 | 61.3 | 74.8 | 74.7 |
| | DoRA | 56.5M | 68.0 | 80.6 | 77.9 | **83.9** | 80.8 | 81.4 | 63.4 | 77.6 | **76.7** |
| | **FoRA** | **9.60M** | 67.8 | 80.1 | 77.5 | 76.6 | 79.8 | 80.3 | 62.8 | 75.2 | 75.0 |
| | **DFoRA** | **10.5M** | 68.8 | **81.2** | 78.0 | 81.3 | 79.2 | 78.9 | 63.1 | **79.6** | 76.3 |
| LLaMA$_{13B}$ | Adpt$^H$ | 206M | 71.8 | 83.0 | 79.2 | 88.1 | 82.4 | 82.5 | 67.3 | 81.8 | 79.5 |
| | Adpt$^R$ | 377M | 72.5 | 84.8 | 79.8 | 92.1 | 84.7 | 84.1 | **71.2** | 82.2 | **81.5** |
| | LoRA | 87.2M | 72.1 | 83.5 | 80.5 | 90.5 | 83.7 | 82.8 | 68.3 | 82.4 | 80.5 |
| | DoRA | 88.6M | 72.4 | **84.9** | 81.5 | 92.4 | 84.2 | 84.2 | 69.6 | **82.8** | **81.5** |
| | **FoRA** | **16.0M** | 72.0 | 84.5 | 80.0 | 91.5 | 83.8 | 83.6 | 70.8 | 82.0 | 81.0 |
| | **DFoRA** | **17.4M** | 71.8 | 84.4 | 81.0 | 91.8 | 84.5 | 84.4 | 70.1 | 81.8 | 81.2 |
| LLaMA2$_{7B}$ | LoRA | 55.7M | 69.8 | 79.9 | 79.5 | 83.6 | 82.5 | 79.8 | 64.7 | 81.0 | 77.6 |
| | DoRA | 56.6M | **71.8** | 83.7 | 76.0 | **89.1** | 82.6 | **83.7** | 68.2 | 82.4 | **79.7** |
| | **FoRA** | **9.60M** | 71.6 | 81.5 | 80.0 | 90.5 | 81.9 | 83.6 | 68.0 | 80.0 | 79.6 |
| | **DFoRA** | **10.5M** | 71.5 | 82.4 | 79.5 | 88.2 | 82.6 | 83.5 | 68.5 | 81.0 | **79.7** |
| LLaMA3$_{8B}$ | LoRA | 56.2M | 70.8 | 85.2 | 79.9 | 91.7 | 84.3 | 84.2 | 71.2 | 79.0 | 80.8 |
| | DoRA | 57.0M | **74.6** | **89.3** | 79.9 | **95.5** | 85.6 | 90.5 | 80.4 | 85.8 | 85.2 |
| | **FoRA** | **9.60M** | 74.0 | 88.7 | 80.0 | 95.2 | **86.2** | 90.4 | 77.8 | 85.0 | 84.7 |
| | **DFoRA** | **10.4M** | 74.5 | 89.1 | 80.4 | 95.1 | 85.8 | 90.6 | 79.7 | 86.8 | 85.3 |

**Results.** Table 3 presents an overview of general performance across different backbone models. Our findings indicate that FoRA consistently outperforms LoRA at the same rank while requiring less than 1/5 parameter count. Furthermore, despite the greater complexity of generalized reasoning tasks, DFoRA either closely matches or even exceeds the performance of DoRA on more advanced LLaMA models, mirroring trends observed in mathematical reasoning. Overall, there is significant variability in the results for commonsense reasoning, with no single method emerging as a definitive leader across all datasets.

## 4.4 IMAGE CLASSIFICATION

This section concentrates on image classification to evaluate whether FoRA can remain competitive. We adopt Vision Transformer (ViT) (Dosovitskiy et al., 2020), which is pre-trained on the vast ImageNet-21K dataset (Ridnik et al., 2021), as the foundation model. Specifically, we fine-tune $ViT_{BASE}$ and $ViT_{LARGE}$ on a variety of datasets, including OxfordPets (Parkhi et al., 2012), Stanford-Cars (Krause et al., 2013), DTD (Cimpoi et al., 2014), EuroSAT (Helber et al., 2019), FGVC (Maji et al., 2013), and RESISC45 (Cheng et al., 2017). Notably, RESISC45 and EuroSAT offer rich labeled data, while the other datasets serve as few-shot adaptations with sparse training samples.

We follow the same fine-tuning protocols as in the GLUE benchmark, reporting the number of trainable parameters excluding the classification head. For LoRA, we set the rank to $r = 16$. To maintain the same parameter constraints, we use $n = 16000$ learnable spectral entries for FourierFT and $n = 8000$ for FoRA. Learning rates are tuned over a maximum of 10 training epochs, and we report average results across 5 random trials. Detailed hyperparameters are provided in Table 9 in the Appendix.

Table 4: Fine-tuning results with $ViT_{BASE}$ and $ViT_{LARGE}$ models on different image classification datasets. Linear Probing (LP) represents fine-tuning only the classification head. Results are averaged across 5 runs with different random seeds. The best performance is shown in **bold**.

| | Methods | # Trainable Parameters | OxfordPets | StanfordCars | DTD | EuroSAT | FGVC | RESISC45 | Avg. |
|---|---|---|---|---|---|---|---|---|---|
| BASE | LP | - | $90.28_{\pm0.43}$ | $25.76_{\pm0.28}$ | $69.77_{\pm0.67}$ | $88.72_{\pm0.13}$ | $17.44_{\pm0.43}$ | $74.22_{\pm0.10}$ | 61.03 |
| | FF | 85.8M | $92.82_{\pm0.54}$ | $\mathbf{85.10}_{\pm0.21}$ | $80.11_{\pm0.56}$ | $\mathbf{99.11}_{\pm0.07}$ | $\mathbf{61.60}_{\pm1.00}$ | $\mathbf{96.00}_{\pm0.23}$ | **85.79** |
| | LoRA | 0.59M | $93.76_{\pm0.44}$ | $78.04_{\pm0.33}$ | $78.56_{\pm0.62}$ | $98.84_{\pm0.08}$ | $56.64_{\pm0.55}$ | $94.66_{\pm0.17}$ | 83.42 |
| | FourierFT | 0.384M | $93.37_{\pm0.30}$ | $81.22_{\pm0.48}$ | $78.90_{\pm0.75}$ | $98.92_{\pm0.09}$ | $58.82_{\pm0.37}$ | $94.91_{\pm0.24}$ | 84.36 |
| | **FoRA** | **0.384M** | $\mathbf{94.05}_{\pm0.37}$ | $81.46_{\pm0.78}$ | $\mathbf{80.34}_{\pm1.03}$ | $98.85_{\pm0.10}$ | $58.67_{\pm0.37}$ | $94.89_{\pm0.15}$ | 84.71 |
| LARGE | LP | - | $91.11_{\pm0.30}$ | $37.91_{\pm0.27}$ | $73.33_{\pm0.26}$ | $92.64_{\pm0.08}$ | $24.62_{\pm0.24}$ | $82.02_{\pm0.11}$ | 66.94 |
| | FF | 303M | $94.30_{\pm0.31}$ | $\mathbf{88.15}_{\pm0.50}$ | $80.18_{\pm0.66}$ | $\mathbf{99.06}_{\pm0.10}$ | $\mathbf{67.38}_{\pm1.06}$ | $\mathbf{96.08}_{\pm0.20}$ | **87.53** |
| | LoRA | 1.57M | $94.62_{\pm0.47}$ | $86.11_{\pm0.42}$ | $80.09_{\pm0.42}$ | $98.99_{\pm0.03}$ | $63.64_{\pm0.83}$ | $95.94_{\pm0.21}$ | 86.56 |
| | FourierFT | 0.768M | $\mathbf{94.91}_{\pm0.33}$ | $85.93_{\pm0.58}$ | $81.17_{\pm0.71}$ | $99.04_{\pm0.07}$ | $62.48_{\pm0.45}$ | $95.59_{\pm0.23}$ | 86.52 |
| | **FoRA** | **0.768M** | $94.90_{\pm0.20}$ | $86.23_{\pm0.29}$ | $\mathbf{81.91}_{\pm0.82}$ | $99.06_{\pm0.09}$ | $65.61_{\pm0.72}$ | $95.81_{\pm0.13}$ | 87.25 |

**Results.** Table 4 presents a comprehensive overview across 6 distinct image classification datasets using $ViT_{BASE}$ and $ViT_{LARGE}$. FoRA consistently outperforms LoRA by a significant margin while using only half the number of trainable parameters. Additionally, FoRA demonstrates superior performance compared to FourierFT under the same parameter constraints. Notably, FoRA even achieves results on par with full fine-tuning, despite utilizing substantially fewer parameters. These findings, along with the insights from Figure 3, highlight the importance of enabling flexible rank adaptation with reduced overhead to enhance representational power.

## 4.5 ANALYSIS

**Sparse Learning Strategy.** To explore the impact of various sparse learning strategies **applied to LoRA**, we compare FoRA with two prominent strategies, random masking (Masking) and linear projection (VeRA) (Kopiczko et al., 2023), assessing their performance compared to LoRA across different tasks and ranks. We fine-tune $RoBERTa_{BASE}$ and $ViT_{BASE}$ on three representative datasets respectively, following the setup in Section 4.1 and 4.4. To ensure fairness, the number of retained parameters for random masking matches the learnable coefficients in FoRA.

The average accuracies across different ranks are depicted in Figure 3, with the corresponding parameter counts detailed in Table 10 in the Appendix. FoRA demonstrates a performance pattern akin to LoRA, closely matching its results across various ranks, particularly at higher ranks, while maintaining a more flexible and reduced parameter count that can be adjusted based on task complexity. In contrast, random masking shows degraded performance compared to FoRA in the GLUE, likely due to the limited expressiveness of trivial masking with extremely sparse parameters. Surprisingly, despite the decent performance in GLUE, VeRA shows a notable drop in more challenging image classification tasks, even when using high-rank matrices. This drop may stem from its inflexible parameter count constrained by the size of the adaptation matrices. Overall, these findings suggest

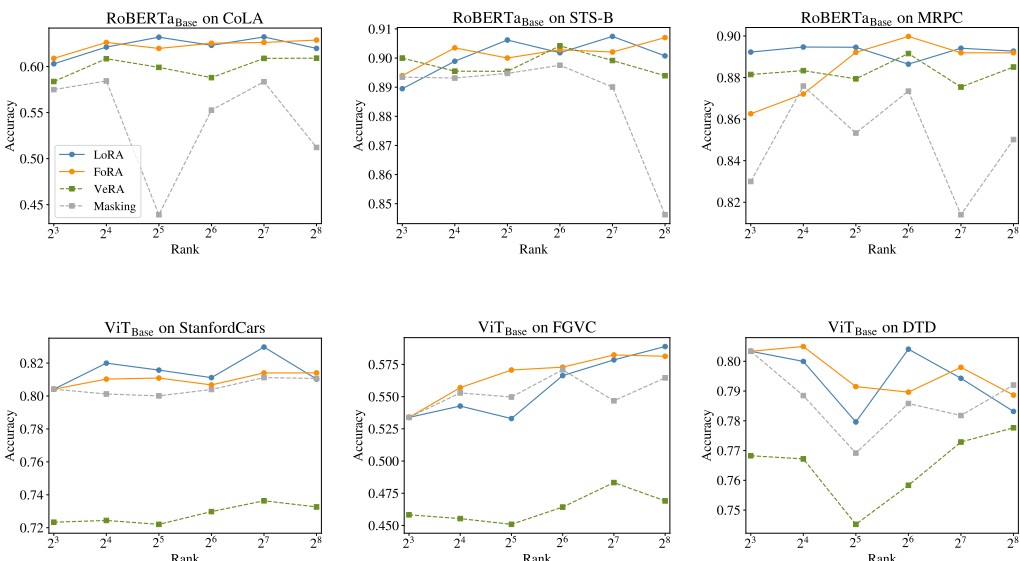

Figure 3: Performance comparison of LoRA variants with different parameter-reduction strategies applied to RoBERTa$_{\text{BASE}}$ and ViT$_{\text{BASE}}$ across various rank configurations. FoRA consistently matches LoRA's performance, while other variants show varying levels of degradation.

that the stronger expressive power of the Fourier basis, combined with the flexible adjustment of trainable parameters, positions FoRA as a promising and parameter-efficient alternative to LoRA.

**Efficiency Comparison.** To assess the computational efficiency, we compare the training time and GPU overhead of FoRA against LoRA for fine-tuning LLaMA2$_{7B}$ on MATH and Comonsense170K, adhering to the setup in Section 4.2 and 4.3. Our evaluation covers both low-rank ($r = 32$) and high-rank ($r = 256$) scenarios to ensure a comprehensive comparison. As shown in Table 5, despite the additional operations introduced by the Fourier transform in FoRA's forward pass, the impact on training time remains modest, with an increase of up to only 4%, even when fine-tuning high-rank, large-scale datasets. Moreover, FoRA demonstrates improved GPU memory efficiency, particularly in high-rank scenarios, reducing memory usage by up to 5.3%. These findings highlight that FoRA also strikes a great balance between memory efficiency and training time

Table 5: Comparison of GPU memory and training time.

| Dataset | Methods | $r = 32$ | | $r = 256$ | |
|---------|---------|--------|------|--------|------|
| | | Memory | Time | Memory | Time |
| MATH | LoRA | 34.9 GB | **37 min** | 37.3 GB | **38 min** |
| | FoRA | **34.4 GB** | 37.5 min | **35.3 GB** | 38.5 min |
| Common | LoRA | 42.4 GB | **442 min** | 45.3 GB | **466 min** |
| | **FoRA** | **41.9 GB** | 454 min | **43.4 GB** | 485 min |

## 5 CONCLUSION

In this work, we aim to unlock the rank-bounded potential of LoRA while minimizing and controlling parameter overhead. We present FoRA, a fine-tuning method that re-parameterizes adaptation matrices from spectral subspace and is compatible with LoRA and its variants. With Fourier basis, FoRA allows for the representation of informative adaptation matrices from lower to potentially unbounded ranks at fixed parameter cost. Empirically, FoRA consistently matches or surpasses LoRA's performance across various fine-tuning tasks and backbone models, requiring up to 15x fewer trainable parameters. Moreover, a comprehensive analysis further substantiates FoRA as a parameter-efficient alternative to LoRA. Our work demonstrates the potential for efficiently replicating LoRA's capabilities, with opportunities for further exploration in future research.

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

# A ADDITIONAL EXPERIMENTAL DETAILS

## A.1 COMPUTATIONAL HARDWARE

All our experiments were carried out on Linux servers equipped with an AMD EPYC 7763 64-Core CPU processor, 512GB RAM, and NVIDIA RTX 6000 ADA 48G / A800 80G GPU.

## A.2 HYPERPARAMETERS

Table 6: Hyperparameter configurations for GLUE benchmark.

| Model | Hyperparameter | SST-2 | MRPC | CoLA | QNLI | RTE | STS-B |
|---|---|---|---|---|---|---|---|
| | Optimizer | | | AdamW | | | |
| | LR Scheduler | | | Linear | | | |
| | Warmup Ratio | | | 0.06 | | | |
| | Max Seq. Len. | | | 512 | | | |
| | Spectral Coefficients $n$ | | | {250,500} | | | |
| BASE | Rank $r$ | 32 | 64 | 64 | 8 | 32 | 256 |
| | Epochs | 50 | 30 | 100 | 40 | 100 | 90 |
| | Batch Size | 128 | 32 | 128 | 32 | 32 | 32 |
| | LR (Head) | 6E-4 | 6E-4 | 3E-4 | 6E-5 | 3E-4 | 2E-4 |
| | LR (FoRA) | 2E-2 | 4E-2 | 4E-2 | 7E-2 | 3E-2 | 2E-2 |
| LARGE | Rank $r$ | 32 | 32 | 32 | 32 | 32 | 32 |
| | Epochs | 20 | 50 | 100 | 30 | 70 | 40 |
| | Batch Size | 128 | 32 | 128 | 8 | 32 | 32 |
| | LR (Head) | 1E-4 | 2E-4 | 4E-4 | 4E-4 | 3E-4 | 7E-5 |
| | LR (FoRA) | 3E-2 | 5E-2 | 4E-2 | 2E-2 | 2E-2 | 3E-2 |

Table 7: Hyperparameter configurations for mathematical reasoning.

| Hyperparameter | LLaMA2$_{7B}$ | | LLaMA2$_{13B}$ | | LLaMA3$_{8B}$ | |
|---|---|---|---|---|---|---|
| | GSM8k | MATH | GSM8k | MATH | GSM8k | MATH |
| Optimizer | | | AdamW | | | |
| LR Scheduler | | | Cosine | | | |
| Batch Size | | | 16 | | | |
| Warmup Ratio | | | 0.05 | | | |
| Dropout | | | 0.05 | | | |
| Epochs | | | 3 | | | |
| Where | | | Q,V | | | |
| Spectral Coefficients $n$ | 20000 | | 30000 | | 20000 | |
| Rank $r$ (FoRA) | 256 | 128 | 256 | 128 | 256 | 128 |
| Rank $r$ (DFoRA) | 256 | 128 | 256 | 128 | 128 | 128 |
| LR (LoRA) | 5E-4 | 5E-4 | 5E-4 | 6E-4 | 5E-4 | 5E-4 |
| LR (DoRA) | 4E-4 | 5E-4 | 4E-4 | 6E-4 | 6E-4 | 2E-4 |
| LR (FoRA) | 6E-3 | 5E-3 | 5E-3 | 5E-3 | 1E-3 | 9E-4 |
| LR (DFoRA) | 5E-3 | 3E-3 | 6E-3 | 6E-3 | 1E-3 | 9E-4 |

Table 8: Hyperparameter configurations for commonsense reasoning.

| Hyperparameter | LLaMA$_{7B}$ | | LLaMA$_{13B}$ | | LLaMA2$_{7B}$ | | LLaMA3$_{8B}$ | |
|---|---|---|---|---|---|---|---|---|
| | FoRA | DFoRA | FoRA | DFoRA | FoRA | DFoRA | FoRA | DFoRA |
| Optimizer | AdamW | | | | | | | |
| LR Scheduler | Linear | | | | | | | |
| Batch Size | 16 | | | | | | | |
| Warmup Steps | 100 | | | | | | | |
| Dropout | 0.05 | | | | | | | |
| Epochs | 3 | | | | | | | |
| Rank $r$ | 32 | | | | | | | |
| Alpha $\alpha$ | 64 | | | | | | | |
| Where | Q,K,V,Up,Down | | | | | | | |
| Spectral Coefficients $n$ | 30000 | | 40000 | | 30000 | | 30000 | |
| LR | 1E-3 | 1.4E-3 | 9E-4 | 9E-4 | 8E-4 | 8E-4 | 5E-4 | 5E-4 |

Table 9: Hyperparameter configurations for finetuning ViT on the image classification datasets.

| Model | Hyperparameter | OxfordPets | StanfordCars | DTD | EuroSAT | FGVC | RESISC |
|---|---|---|---|---|---|---|---|
| | Optimizer | AdamW | | | | | |
| | Epochs | 10 | | | | | |
| | Batch Size | 64 | | | | | |
| | Rank $r$ (LoRA) | 16 | | | | | |
| | Spectral Coefficients $n$ | 8000 | | | | | |
| BASE | Rank $r$ (FoRA) | 32 | 128 | 64 | 64 | 256 | 32 |
| | LR (Head) | 8E-3 | 1E-2 | 1E-2 | 1E-4 | 1E-2 | 1E-2 |
| | LR (FoRA) | 4E-3 | 5E-2 | 5E-3 | 2E-2 | 5E-2 | 2E-2 |
| | Weight Decay | 4E-2 | 1E-5 | 2E-4 | 4E-3 | 2E-2 | 9E-2 |
| LARGE | Rank $r$ (FoRA) | 64 | 128 | 128 | 64 | 256 | 32 |
| | LR (Head) | 6E-3 | 5E-3 | 1E-2 | 1E-3 | 1E-2 | 1E-2 |
| | LR (FoRA) | 5E-3 | 3E-2 | 4E-3 | 3E-2 | 8E-2 | 1E-2 |
| | Weight Decay | 3E-4 | 2E-5 | 3E-5 | 3E-3 | 1E-2 | 1E-3 |

## A.3 PARAMETER COUNT OF SPARSE LEARNING STRATEGIES

As the rank increases, the number of learnable parameters in LoRA grows linearly, leading to a significant parameter overhead. While VeRA exhibits a minimal increase in parameters, its strong dependence on the size of its adaptation matrices limits its flexibility in adapting to more complex tasks. In contrast, both FoRA and random masking maintain a fixed number of learnable parameters across different ranks, providing greater flexibility by allowing parameter adjustments based on task complexity.

Table 10: Comparison of learnable parameters across different compression strategies.

|  | Methods | Rank $r$ | | | | | |
|---|---|---|---|---|---|---|---|
|  |  | $2^3$ | $2^4$ | $2^5$ | $2^6$ | $2^7$ | $2^8$ |
| RoBERTa$_{\text{BASE}}$ | LoRA | 6,144 | 12,288 | 24,576 | 49,152 | 98,304 | 196,608 |
|  | VeRA | 776 | 784 | 800 | 832 | 896 | 1024 |
|  | FoRA/Mask | 500 | 500 | 500 | 500 | 500 | 500 |
| ViT$_{\text{BASE}}$ | LoRA | 6,144 | 12,288 | 24,576 | 49,152 | 98,304 | 196,608 |
|  | VeRA | 776 | 784 | 800 | 832 | 896 | 1024 |
|  | FoRA/Mask | 6,144 | 8000 | 8000 | 8000 | 8000 | 8000 |

# B IMPLEMETATION

Algorithm 1 presents the PyTorch implementation of FoRA. Our approach allows for a straight-forward plug-in, with the only modification needed being the replacement of the PyTorch linear module with the FoRA linear module. Additionally, we create a cached empty spectral matrix that matches the size of the adaptation matrix. This caching strategy not only accelerates GPU computations but also minimizes GPU overhead, as only the sparse spectral coefficients require backward propagation.

---

**Algorithm 1** FoRA PyTorch Implementation

---

```python
class PEFT(nn.Module):
    def __init__(self, ...):
        if LoRA:
            A, B = torch.nn.Linear(d1, r), torch.nn.Linear(r, d2)
        elif FoRA:
            A, B = FoRALinear(n, d1, r), FoRALinear(n, r, d2)

class FoRALinear(nn.Module):
    def __init__(self, n, din, dout):
        self.n = n # number of spectral coefficients
        self.din = din # input dimension
        self.dout = dout # output dimension
        # spectral location
        self.L = torch.randperm(din * dout)[:n]
        self.L = torch.stack([self.L // din, self.L % dout], dim=0)
        # spectral coefficients
        self.s = nn.Parameter(torch.zeros(n))
        # cached spectral matrix
        self.cached_F = torch.zeros(self.din, self.dout)

    def weight(self):
        # spectral assignment
        F = self.cached_F.clone()
        F[self.L[0], self.L[1]] = self.s
        # spatial conversion
        W = torch.fft.ifft2(F).real()
        return W
```

---

## C   GENERATION EXAMPLES

---

**Instruction Tuning (MATH)**

How many inches are in the circumference of a great circle of a sphere which has a surface area of $196\pi$ sq in? Express your answer in terms of $\pi$.
Note: To measure the circumference, take the largest circle on the surface of the sphere.
Answer the above question. First think step by step and then answer the final number.

**Response**

| Model | Response |
|---|---|
| **LLaMA2$_{7B}$ (FoRA)** | The surface area of a sphere is $\pi r^2$, where $r$ is the radius. So, the radius is $r = \sqrt{196\pi} = 12\sqrt{3}$. The circumference of the sphere is $2\pi r = \boxed{24\pi}$. The final answer is: $24\pi$. |
| **LLaMA3$_{8B}$ (FoRA)** | The surface area of a sphere is $4\pi r^2$, so $r^2 = 49$ and $r = 7$. The circumference of the great circle is $2\pi r = \boxed{14\pi}$. The final answer is: $14\pi$. |

---

**Instruction Tuning (HellaS.)**

Please choose the correct ending to complete the given sentence: Roof shingle removal: A man is sitting on a roof. he
Ending1: is using wrap to wrap a pair of skis. Ending2: is ripping level tiles off. Ending3: is holding a rubik's cube. Ending4: starts pulling up roofing on a roof.
Answer format: ending1/ending2/ending3/ending4

**Response**

| Model | Response |
|---|---|
| **LLaMA2$_{7B}$ (FoRA)** | the correct answer is ending2 |
| **LLaMA3$_{8B}$ (FoRA)** | the correct answer is ending4 |

