# OpenReview forum: "LoRA Unleashed: Effortlessly Advancing from Low to Arbitrary Rank"
_ICLR.cc/2025/Conference — ICLR 2025 Conference Withdrawn Submission_

### Official Review · Reviewer_46jZ · 2024-10-28

**Soundness:** 1
**Presentation:** 1
**Contribution:** 2
**Rating:** 3
**Confidence:** 4

**Summary:**

For fine-tuning foundation models, the authors propose a variant of low rank approximation where each factor is the Fourier transform of a sparse weight matrix whose sparsity pattern is random.

**Strengths:**

According to the authors' experimental results, the proposed FoRA gives decent performance with few trainable parameters.

**Weaknesses:**

The intellectual contribution is lacking.
* The authors provide no valid explanation/intuition for why the proposed method should work well.  The citations to Candes and Baraniuk on line 180 are invalid because those works fit the sparsity pattern to the data, whereas the authors choose it randomly.
* The suggestion on line 204, that FoRA has an advantage over LoRA because it can construct a high rank gradient approximation with few parameters, is mathematically flawed. The authors are forgetting that LoRA can learn ANY rank-r approximation while FoRA can learn only a SUBSET of rank-r approximations. The authors give no explanation of why FoRA's subset is a good one.
* Around line 218 the authors criticize existing work on linear projections of LoRA. But that is exactly what the authors are proposing. Their Atilde and Btilde are linearly parametrized by the s coefficients and thus linear projections of the LoRA A and B matrices.
* The competing FourierFT method sounds very similar to the proposed method. Why is FourierFT never discussed? This omission brings up questions of novelty.

The description of the proposed method is highly problematic
* What exactly are the contents of the index matrix L? If integers, what is their range? Can they repeat?
* Above (3) it’s said that F is r1 by d. What s r1? What is d?
* If F is r1 by d, then the Fourier transform (3) is wrong: the denominator coefficients in the exponential must be r1 and d, not r and d2
* If F is r1 by d, then this conflicts with Figure 2, which states that F is d1 by r
* Both the stated dimensions of F and Atilde above (3) and in Figure 2 conflict with (2), which says they are r by d2
* It is said after (4) that Btilde is constructed using the “identical” procedure, but this makes no sense given that Btilde has different dimensions than Atilde
* On line 213, the “spatial auxiliary matrix” A’ is never defined
* The mathematical notation in the paper is extremely confusing.  In some cases, capital boldface quantities with subscripts (e.g., $\mathbf{W}_0$) denote matrices, but in other cases (e.g., $\mathbf{F}\_{j,k}, \mathbf{S}\_{p,q}$) they denote scalars.

Many claims about the proposed method simply do not hold
* The claim (line 82) that A and B can have “unbounded ranks” is mathematically false, since it is clear from (2) that the rank is at most r. Furthermore, the rank of the proposed method in (2) is no different than the standard LoRA method (1), which is at most r.
* In many places the authors claim they are “maximizing “ potential while “minimizing “ parameter overhead, and achieving the “optimal” balance of efficiency and performance. But there are no optimization problems and no evidence of optimality in the paper.

The experimental results are not convincing
* Figure 1, which supposedly shows the performance of LoRA versus rank, doesn’t support the authors claims (line 43) that LoRA performance increases with rank. Rather, the performance seems almost totally random. Thus the entire motivation for the paper falls apart.
* In Table 1, the proposed FoRA wins in only 3 of the 14 contests!
* The experiment descriptions are missing key information, such as the dimensions d1 and d2, and in some cases (e.g ., GLUE) the FoRA rank r. Without this information, it’s difficult to understand the degree of the approximation and the meaning of n.

**Questions:**

I have no questions.

---

### Official Review · Reviewer_bQVc · 2024-10-28

**Soundness:** 2
**Presentation:** 3
**Contribution:** 2
**Rating:** 5
**Confidence:** 4

**Summary:**

This paper introduces Fourier-based Flexible Rank Adaptation, an approach that re-parameterizes adaptation matrices through a sparse spectral subspace to enhance expressiveness.

**Strengths:**

The paper is well-structured and presented in a clear, reader-friendly manner.

**Weaknesses:**

1. **Contradictory Claims in Abstract:**
   The claims in the abstract (line 11) present a potentially contradictory view. The authors commend Low-Rank Adaptation (LoRA) for reducing parameters through low-rank matrices (∆W = BA) but then criticize the method's limited expressiveness due to the same low-rank structure. Clarification would be beneficial as to whether expressiveness is indeed a key metric in evaluating LoRA’s effectiveness. Additionally, since LoRA's rank is adjustable, it is unclear why the method could not meet various levels of expressiveness, which calls into question the necessity of the proposed study.

2. **Rank Discrepancy in FoRA and LoRA (Figure 2):**
   In Figure 2, the rank parameter (r’) in FoRA appears much larger than the rank parameter (r) in LoRA, which raises questions. Since a higher rank implies a broader span in the basis space, comparing LoRA and FoRA with different ranks seems misleading. Furthermore, in lines 152 and 182, LoRA and FoRA appear to be assigned the same rank (r), which adds to the confusion. Greater clarity on this point would strengthen the paper.

3. **Incremental Novelty:**
   The proposed method introduces the Fast Fourier Transform to augment classic LoRA, which, while effective, may represent an incremental rather than important advancement.

4. **Lack of Specific Motivation and Details:**
   The paper frequently uses broad claims regarding the capabilities of the approach but lacks specific details on the motivation and mechanisms that enable these achievements. More detailed and concrete explanations would be beneficial.

5. **Computational Complexity:**
   Table 1 indicates modest performance gains; however, these benefits may be offset by the increased computational complexity introduced by re-parameterizing A and B in LoRA.

**Questions:**

- Could the authors provide additional details on Equations (2) and (3)?
- Is FoRA a two-layer low-rank representation?

---

### Official Review · Reviewer_Qa4z · 2024-11-01

**Soundness:** 2
**Presentation:** 3
**Contribution:** 2
**Rating:** 5
**Confidence:** 3

**Summary:**

This paper proposed a new variant of LoRA, called FoRA, by virtue of Fourier bases representation, to reduce the parameters to be fine-tuned. The core idea is to further reparameterize the adaptation matrices using Fourier bases. Due to the powerful representation ability of the Fourier bases, the tunable parameters can be much less than that of the original adaptation matrices, while still maintaining the fine-tuning efficacy. Experiments on several different tasks show the promising performance of FoRA.

**Strengths:**

1. As verified by the experiments, the trainable parameters of FoFA are indeed much fewer than LoRA to achieve a comparable or better performance.

2. Like LoRA, the proposed method is a generally available fine-tuning method for large foundation models on various tasks.

**Weaknesses:**

My major concern is about the computational efficiency. To my knowledge, the ultimate goal of LoRA-type methods is to reduce the computational cost for fine-tuning the large models. However, as shown in Table 5 of the manuscript, though the trainable parameters have been significantly reduced, it seems that both the memory and time costs are not better than LoRA. Therefore, I do not believe the proposed method is more meaningful than LoRA in applications.

**Questions:**

Since the computational efficiency of the proposed method reflected in Table 5 is impressive, the authors should discuss and demonstrate other benefits that the proposed method can bring over LoRA.

---

### Official Review · Reviewer_E6MC · 2024-11-04

**Soundness:** 2
**Presentation:** 2
**Contribution:** 1
**Rating:** 3
**Confidence:** 5

**Summary:**

This work introduces a method called Fourier-based Flexible Rank Adaptation (FoRA) to enhance Low-Rank Adaptation (LoRA) for fine-tuning large foundation models. However, this approach can be seen as a straightforward combination of LoRA and FourierFT, simply applying FourierFT within the LoRA framework. Therefore, I believe the contribution of this work is quite limited.

**Strengths:**

The authors provides the details of the experiments along with the code, which is highly commendable.

**Weaknesses:**

(1) The paper lacks discussion of some key related works, such as AdaLoRA, and does not include comparisons with them in the experiments. AdaLoRA also addresses, to some extent, the issue mentioned in the second paragraph of the paper, where "different tasks exhibit varying sensitivity to rank."
(2) This work appears to be a straightforward combination of LoRA and Fourier methods, applying FourierFT within the LoRA framework.
(3) The experimental results of FoRA show no significant advantage in terms of parameter count or accuracy compared to other methods, such as FourierFT.

**Questions:**

See the above.

---

### Note · Authors · 2024-11-26

I have read and agree with the venue's withdrawal policy on behalf of myself and my co-authors.